# Cell-Type-Dependent Recruitment Dynamics of FUS Protein at Laser-Induced DNA Damage Sites

**DOI:** 10.3390/ijms25063526

**Published:** 2024-03-20

**Authors:** Yu Niu, Arun Pal, Barbara Szewczyk, Julia Japtok, Marcel Naumann, Hannes Glaß, Andreas Hermann

**Affiliations:** 1Department of Neurology, Technische Universität Dresden, 01307 Dresden, Germany; yuniu1215@yahoo.com (Y.N.); a.pal@hzdr.de (A.P.); julia.japtok@ukdd.de (J.J.); 2Dresden High Magnetic Field Laboratory (HLD), Helmholtz-Zentrum Dresden-Rossendorf (HZDR), 01328 Dresden, Germany; 3Translational Neurodegeneration Section “Albrecht Kossel”, Department of Neurology, University Medical Center Rostock, University of Rostock, 18147 Rostock, Germany; szewczyk@mpi-cbg.de (B.S.); marcel.naumann@med.uni-rostock.de (M.N.); hannes.glass@med.uni-rostock.de (H.G.); 4German Center for Neurodegenerative Diseases (DZNE), Site Rostock/Greifswald, 18147 Rostock, Germany; 5Center for Transdisciplinary Neurosciences Rostock (CTNR), University Medical Center Rostock, University of Rostock, 18147 Rostock, Germany

**Keywords:** cell type, DNA damage, FUS, human induced pluripotent stem cells, laser micro irradiation, kinetics

## Abstract

Increased signs of DNA damage have been associated to aging and neurodegenerative diseases. DNA damage repair mechanisms are tightly regulated and involve different pathways depending on cell types and proliferative vs. postmitotic states. Amongst them, fused in sarcoma (FUS) was reported to be involved in different pathways of single- and double-strand break repair, including an early recruitment to DNA damage. FUS is a ubiquitously expressed protein, but if mutated, leads to a more or less selective motor neurodegeneration, causing amyotrophic lateral sclerosis (ALS). Of note, ALS-causing mutation leads to impaired DNA damage repair. We thus asked whether FUS recruitment dynamics differ across different cell types putatively contributing to such cell-type-specific vulnerability. For this, we generated engineered human induced pluripotent stem cells carrying wild-type FUS-eGFP and analyzed different derivatives from these, combining a laser micro-irradiation technique and a workflow to analyze the real-time process of FUS at DNA damage sites. All cells showed FUS recruitment to DNA damage sites except for hiPSC, with only 70% of cells recruiting FUS. In-depth analysis of the kinetics of FUS recruitment at DNA damage sites revealed differences among cellular types in response to laser-irradiation-induced DNA damage. Our work suggests a cell-type-dependent recruitment behavior of FUS during the DNA damage response and repair procedure. The presented workflow might be a valuable tool for studying the proteins recruited at the DNA damage site in a real-time course.

## 1. Introduction

Tissue or even cell type selectivity is central in many different disease conditions, including cancer but also neurodegenerative diseases. In the latter, despite carrying germline mutations, a more or less selective degeneration of neuroectodermal or even specific neuronal subpopulations appears [1]. Interestingly, while in many neurodegenerative diseases signs of increased DNA damage are reported, some neurodegenerative diseases are caused by mutations in genes known to be involved directly in the DNA damage response (DDR) pathway. Examples of these are Ataxia teleangiectatica—caused by autosomal recessive mutations in the *ATAXIA TELEANGIECTASIA MUTATED* (*ATM*) gene—or Amyotrophic lateral sclerosis (ALS)—caused, among others, by autosomal dominant mutations within the *FUSED IN SARCOMA* (*FUS*) gene. Mutations in *ATM* increase the risk of cancer and the radiation sensitivity of various somatic cell populations despite the selective degeneration of cerebellar neurons, while mutations in *FUS* solely affect motor neurons without signs of an increased cancer risk [2].

FUS, also known as translocated in liposarcoma (TLS), is a member of the TET (TAF15, EWS, and TLS) family of RNA-binding proteins [3]. Although first described as cancer-related, FUS was later found to be a ubiquitously expressed protein [4]. It is a multi-domain heterogeneous nuclear RNA-binding protein (hnRNP) [3] with multiple biological functions such as involvement in several steps of RNA metabolism including transcription, splicing, RNA transport, and translation [3,5,6]. In addition, FUS was reported to be one of the early recruited proteins at the laser irradiation-induced DNA damage site (DDS), where it participates in DDR and DNA repair. In this respect, FUS was reported to play a role in the two major pathways of double-strand break (DSB) repair, namely in the homologous recombination (HR) and non-homologous end joining (NHEJ) [6,7,8]. For example, it participates in the HR by promoting the formation of DNA D-loops and annealing homologous DNA [9,10]. FUS has a high affinity with double-stranded and single-stranded DNA [11]. In line with this, recent data also point towards a role of FUS in single-strand break (SSB) repair by promoting adherence of the X-ray repair cross-complementing protein 1 (XRCC1)/ DNA ligase 3 (LIG3) complex to the SSB site downstream of Poly-ADP-ribose (PAR) polymerase 1 (PARP1) [7]. FUS was also shown to directly interact with PARP1 and HDAC1 protein at the site of DNA damage [8,12], and, in response to DNA damage, was phosphorylated by ATM [13] and DNA-PK [14].

Even though FUS has been extensively studied concerning its biochemical mechanisms and interactions with other proteins of the DDR machinery, little is known about its tissue selectivity or impact on the dynamics of DNA repair pathways in different cell types. This is further emphasized, since mutations within *FUS* cause a selective motor neuron degeneration despite the protein being ubiquitously expressed [15]. 

In order to shed light on this question, we used a combination of a pulsed UV laser micro-irradiation (IR), which is known to generate oxidative DNA damage, and live cell microscopy, together as a well-established and powerful tool to analyze spatiotemporally defined induced DNA damage [16,17,18]. Monitoring a variety of cells expressing eGFP-tagged FUS enabled us to assess the recruitment kinetics of this key DNA repair protein in different cell types live [19].

Induced pluripotent stem cells (iPSCs) were first described in 2006 and only one year later in 2007, were adopted to human cells. iPSCs can be used to generate unlimited patient-specific cells of nearly all somatic cell types [20]. This technique not only allows for the in vitro culture of hard-to-obtain cell types (such as patient-derived neurons) but also allows us to study the differential vulnerability of various cell types from the same patient (i.e., isogenic conditions). This is of high interest because many diseases are caused by specific mutations in genes, which are expressed in several cell types but only show selective degeneration (e.g., motor neurons in the case of ALS, or striatal neurons in the case of Huntingon’s disease). 

In this study, we generated iPSC lines in which endogenous wild-type FUS is c-terminally tagged eGFP to visualize FUS during the early response to UV laser micro irradiation-induced DNA damage in various cell types. We additionally developed a robust workflow, to measure the kinetics of FUS protein recruitment to and dissociation from the DDS, similar to what was reported very recently [21,22,23]. Our obtained multiparametric results describe the dynamics of FUS kinetics at laser-induced DDS in a quantitative and comprehensive manner. We found cell-type-dependent kinetics of FUS in response to DNA damage. Our analytical workflow is also suitable for analyzing the dynamics of other DDS proteins and thus for addressing a broad spectrum of neurodegenerative and cancer biology questions.

## 2. Results

### 2.1. Generation of iPSC Lines Expressing Endogenously Tagged FUS-eGFP

In order to investigate the recruitment dynamics of FUS in response to UV laser-induced DNA damage, we used CRISPR/Cas9n to generate hiPSC lines in which endogenous wild-type FUS is tagged with a c-terminal eGFP (Figure 1a). We transfected healthy donor-derived hiPSC lines following a previously established protocol from our group. After one to two weeks of selection (details in material and methods part), eGFP-positive hiPSCs were emerging (Figure 1b). We prepared immunofluorescence stainings and genotyping PCRs to confirm the success of the gene editing. Mainly nuclear FUS-eGFP signal showed a strong overlap with FUS- and GFP antibody stainings (Figure 1c). Genotyping PCR confirmed the heterozygosity of the engineered FUS locus (Figure 1d, for the entire gel see Appendix A). Western blot revealed two bands for endogenous FUS as well as FSU-eGFP expression, highlighting the heterozygosity of the established lines (Figure 1e, Appendix A).

### 2.2. All Cells Show FUS Recruitment after UV Laser Irradiation-Induced DDS except a Fraction of Undifferentiated hiPSC

The reasons for tissue or even cell type selectivity in neurodegenerative diseases are often unknown [15]. FUS-ALS causing mutations impair DDR, including proper FUS recruitment to UV laser-induced DDS. However, whether this defect is cell-type-specific or not is unknown. Thus, we asked ourselves whether the recruitment dynamics of wild-type FUS are different across various tissues and cell types, thereby shedding some light on the selective vulnerability when being ubiquitously affected.

UV laser micro-irradiation with the same technical settings was thus conducted on six different in vitro cell models. These included five isogenic conditions: gene-edited undifferentiated hiPSCs and their derivatives, which were either meso/endodermal cells or neuroectodermal in the form of small-molecule neuronal progenitor cells (smNPCs). Furthermore, we analyzed differentiated striatal neurons and spinal motor neurons (Figure 2a). We used well-established differentiation protocols for each [24,25,26]; representative marker expression of different cell types is shown in Appendix A. Independent of these hiPSC derivatives, we also included gene-edited HeLa cells expressing wild-type FUS-eGFP through a BAC [27,28]. FUS-eGFP was expressed in the cell nucleus in all cells investigated (Figure 2b). FUS recruitment to the UV laser irradiation-induced DDS was seen in motor neurons, as previously reported. There was no difference to other neuronal subtypes such as striatal interneurons (Figure 2b,c). Interestingly, a fraction of undifferentiated hiPSC did not recruit FUS to DDS within the investigated timeframe (10 min), despite showing high FUS-eGFP expression. This was not due to the difference between postmitotic and cycling cells, since highly proliferative smNPCs did show FUS recruitment to laser-induced DDS in all cells (Figure 2b,c). Furthermore, there was also no difference compared to cells of other germ layers, as shown in endoderm/mesoderm progenitor cells or HeLa cells (Figure 2b,c). The mean fractions of FUS-eGFP recruitment at the DDS in each cell model were calculated, with hiPSC showing recruitment in about 72% ± 7, and the others were 100% ± 0.

### 2.3. Development of a Workflow for Unbiased Analysis of Kinetic of DDR Protein Recruitment Dynamics at Laser-Induced DDSs

We next asked whether the kinetics of FUS engagement at the DDS differ depending on the cell type. For in-depth analysis of the kinetics of FUS-eGFP at the laser-induced DDS, we generated a robust workflow consisting of a novel Fiji macro and R script. Based on the established laser irradiation assay and live cell imaging acquisition technique, the obtained laser irradiation videos were processed with Fiji. By imaging Z-stacks, we accounted for axial movement or focus shift during the acquisition. In accordance with recently published protocols, we also corrected the image for photo bleaching [21,22,23]. Additionally we corrected for lateral movement of cells. Our script automatically detects laser-irradiated sites to which FUS-eGFP was recruited (Figure 3a,b) as well as their respective background (Figure 3c). Next, the average intensity value of the recruited FUS-GFP of these regions of interest were calculated for each time point. Using KNIME, a background-corrected mean intensity value at the DDS was calculated for each time point and each laser-irradiated cell.

For each FUS-eGFP-recruiting DDS, a graph of intensity change over time was generated (Figure 4). From these graphs, we observed that after the laser irradiation, FUS-eGFP was recruited with a delay of several seconds (Figure 4b) followed by a fast increase in fluorescence intensity until the maximum was reached. In some cells, there was a steady state at maximum observed. After that, FUS-eGFP began to dissociate from the DDS. According to this, we defined four phases of FUS-eGFP recruitment (Figure 4c). The crucial start and end points of every phase were automatically recognized by a custom-tailored R-script. Determination of these points and the mean fluorescence value of each phase allowed the algorithm to determine the crude shape of the FUS-eGFP recruitment. Furthermore, the parameters of our recruitment and dissociation model were estimated by robust non-linear regression, and a modeled graph representing this information for each laser-irradiated cell was plotted over the experimental data (Figure 4c–e). This algorithm enabled us to have qualitative characteristics of the FUS-eGFP recruitment at DNA damage sites:**I** recruitment lag phase**II** association phase**III** plateau phase**IV** dissociation phase

Crucial time points are:**A** laser irradiation applied on the nucleus (always at time point of 1 s)**B** FUS-eGFP is recruited to the DDS**C** end of FUS-eGFP recruitment to DDS**D** FUS-eGFP starts to dissociate from the laser irradiating site**E** FUS-eGFP fluorescence returns to a pre-irradiation-like end point level

Thus, nine parameters were analyzed. 

Recruiting lag time: **A** to **B**Association time: **B** to **C**Plateau time: **C** to **D**Dissociation time: **D** to **E** (if E is not within the measured time window, it is extrapolated as the time point, when 95% of the dissociation would have occurred)FUS duration time: **B** to **E (II + III + IV)***K_on_* value: association constant*K_off_* value: dissociation constantFluorescence intensity fold change at laser irradiating site: the maximum intensity of laser irradiating site subtracted by the intensity before irradiation, then divided by the intensity before irradiation.
$$Foldchange=\frac{Intensity_{C}-Intensity_{A}}{Intensity_{A}}$$Plateaued fraction: the percentage of the number of cells, which had a plateau phase compared to all cells that showed FUS recruitment.

### 2.4. FUS-eGFP Recruitment Kinetics Vary in Different Cell Models

After investigating the overall characteristics of FUS-eGFP recruitment to DDS, we wanted to analyze the kinetics of FUS-eGFP in response to UV laser-induced DNA damage in different in vitro cellular models in more detail. For this, we compared the parameters among various cell models (Figure 5a). One-way ANOVA analysis followed by Tukey post hoc tests were applied as a statistical method. It showed several significant differences of the parameters among different types of cells, suggesting that the recruitment kinetics of FUS-eGFP vary in different cell models, with a particularly long dissociation time and reduced plateau fraction in the case of MNs. 

To strengthen the hypothesis of cell-type-specific DDR kinetics, we validated these findings in isogenic sets of experiments using Line Kolf Cas 9+-hiPSC, Line Kolf Cas 9+-NPC and Line Kolf Cas 9+-motor neuron lines (Figure 5b). Since the Line Kolf Cas 9+-NPC and Line Kolf Cas 9+-motor neuron were both derived from Line Kolf Cas 9+-hiPSC (Figure 2a), the differences of these parameters further indicated the cell type dependence of wild-type FUS protein performance in DDR. Although it was difficult to find a common denominator of FUS-eGPF recruitment kinetics in each cell type, we found that variations in kinetics were mainly during the plateau phase and the dissociation phase. In all types of cells which recruit FUS-eGFP after UV laser irradiation, it took about 12.77 s (SEM = 0.5514, *n* = 36) for irradiated nuclei to start recruiting FUS-eGFP to DDS. The association phase was about 65.56 s (SEM = 2.929, *n* = 36) long. Interestingly, the association time was almost one order of magnitude faster than the dissociation time and similar along all models, while the dissociation had higher variability among the models. There was no significant difference in intensity change at the laser irradiating site, which suggests that the amount of FUS-eGFP recruited to the DDS was similar in all cell types. MNs showed a longer dissociation time, reduced *K_off_* value and reduced plateau fraction.

## 3. Discussion

### 3.1. A Workflow to Dissect the Recruitment Kinetics of FUS-eGFP in Response to DNA Damage

In this study, we used a combination of UV-A laser (355 nm) micro-irradiation to induce DNA damage and live-cell imaging to visualize DDR in different cell models with stable, endogenous expression of eGFP-tagged wild-type FUS protein. We developed a workflow to characterize and model the recruitment kinetics of FUS-eGFP in response to laser-induced DNA damage in more detail, which is technically based on similar recent reports [21,22,23]. However, we additionally aimed to develop an openly available resource for semi-automated analysis. DDR is mediated by an extremely complex network comprising numerous proteins and multiple pathways, which are tightly orchestrated [16,29,30]. It remains a challenge to identify the exact spatiotemporal relationship between the components of the DNA damage repair pathways. The precise kinetic timescales of functional proteins at the DDS are of great interest to understand the mechanisms in DNA damage repair both under physiological and pathological conditions better. The technique and workflow used here is a versatile, publicly available analytical profiling tool for studying the crosstalk between components or pathways in the course of complex DNA repair.

One of the key points is the visualization of the protein of interest (POI). It could be a hurdle of the study since the procedure of labeling POI should be taken into consideration. In our study, we labeled endogenous wild-type FUS protein—a key member of DDR machinery—with eGFP. It was reported to robustly maintain its correct folding within a broad range of biological conditions such as relevant temperature and pH ranges, thereby remaining fluorescent without quenching in sub-cellular compartments, and to have hardly any impact on the native function of most proteins [31,32,33]. It is, however, of note that eGFP is relatively large, and thus can potentially impact the POI’s physical and functional characteristics as well as localization. Specifically, FUS is a protein shuttling between the nucleus and cytosol, and a large tag like eGFP may alter this shuttling behavior and, therefore, contribute to or ameliorate one of FUS’ pathological hallmarks, namely the cytoplasmic mislocalization.

Depending on the laser wavelength, different types of lesions are induced by the laser microbeams, including oxidative base lesions [34], single-strand breaks (SSBs), double-strand breaks (DSBs) and photo byproducts [17,34,35,36,37,38,39]. The UV-C (~260 nm) wavelengths were reported to induce selectively UV photoproducts and direct DNA excitation [34,37]. UV-B (290–320 nm) and UV-A (320–400 nm) wavelengths were found to induce mixtures of DNA base lesion, SSBs and DSBs [34]. Additionally, exogenous photosensitizers such as bromodeoxyuridine (BrdU) and Hoechst33258 were also used for assisting the induction of DNA damage [38]. Thus, under most laser irradiation conditions, a heterogenous spectrum of different kinds of lesions depending on wavelength, laser power and chemical composition is created, which might account for limitations of this technique because the evoked DDR could involve multiple repair pathways, thereby hampering specific pathway studies. In our study, a wavelength of 355 nm within the UV-A range was used, which knowingly generates a heterogeneous mix of SSBs and DSBs [16].

The herein presented workflow for image acquisition, detection of nuclei and DDS is a robust tool for processing UV laser ablation experiments. In accordance with other recently published laser micro-irradiation protocols and guidelines, we conducted photo-bleaching correction [21,22,23] and acquired images with a high framerate to resolve the rapid recruitment dynamics of our protein of interest. Additionally, we imaged Z-stacks to account for omni-directional recruitment of FUS, focus drift as well as axial movement of the region of interest (ROI), a problem which can be encountered during laser micro-irradiations [21]. Our algorithm conducts stack registration to compensate lateral movement, which allows for the acquisition of videos over a longer time period, which has been a major limitation in the study of long recruitment events [23]. The detection of nuclei and DDS relies on published thresholding algorithms, which allow for the unbiased selection of ROIs, all available in the free tool Fiji. Using a maximum intensity projection for detection of ROIs enables pixel-accurate analysis of the DDS and using the peri-irradiated region for background correction. While various guidelines are available, we share our macro and script so that other investigators can reproducibly analyze their data without bias. One drawback might be a potential overfitting towards our imaging setup. For different microscopes, fluorophores and recruitment kinetics, adjustments of the thresholding algorithms as well as the detection of recruitment phases will have to be carried out. 

### 3.2. Live Real-Time Kinetics of the DDR Protein FUS at the DDS In Vitro

FUS is known to be involved in the poly(ADP-ribose) polymerase 1 (PARP1)-dependent DNA damage-sensing pathway in the cellular response to laser-induced DNA damage [12,40]. Thus, we used fluorescently labeled, endogenously expressed wild-type FUS to investigate the recruitment kinetics DDS. In all investigated cell models, we observed that FUS-eGFP mainly localized to the nucleus and could quickly be recruited to the DDS upon laser irradiation. This is consistent with reports on other cell models documenting that FUS is predominantly detected in the nucleus [41], and is amongst the very first recruited proteins at the DDS [8,12,16].

So far, FUS is known to participate in DNA repair particularly in DSB damage response and repair, and the upstream key role it plays in the DDR and DNA repair has been widely studied. It is involved in the formation of D-loops, an essential step in homologous recombination [9]. FUS’ activity in DDR is also controlled by phosphorylation either through ATM [13] or DNA-PK [14], which are the two major components in DSB response and repair pathways. FUS was also reported to be rapidly recruited to laser-induced DNA lesions in a PARP-dependent manner and was able to interact with poly-ADP-ribose chains directly [12,40]. In addition, FUS was demonstrated to play a crucial role in the neuronal DDR by direct interaction with histone deacetylase 1 (HDAC1) [8], a chromatin-modifying enzyme which is required for the DDR, particularly through NHEJ [42]. Thus, all these results have implied that FUS is pivotal in DNA repair. Interestingly, however, despite FUS being expressed in every cell type of the body, mutations in FUS cause a selective motor neuron degeneration. This raises the question of whether the DDR response of FUS is cell-type-specific.

Intriguingly, during the study, we found that in all cell types investigated, FUS was always recruited to laser-irradiated nuclear spots except for undifferentiated hiPSCs, in which only about 72% of all irradiated cells recruited FUS to the DDS. This was not dependent on a proliferation/post mitotic state, since the other three proliferating cell lines—meso/endoderm layer cells, NPCs and Hela cells—all showed FUS recruitment to DDSs induced by laser irradiation in a similar fashion as the post mitotic neuronal cells. This strongly suggested that the FUS recruitment phenotype in hiPSC was not due to the proliferative property. Furthermore, the cells derived from hiPSCs all showed FUS recruitment, indicating the DNA damage response and repair mechanism was obviously undergoing alterations during the cells’ differentiation.

Self-renewal is a unique property of stem cells; thus, having a robust DDR and DNA damage repair program to protect their genomic integrity from all kinds of endogenous and exogenous DNA damages is particularly crucial for stem cells. Otherwise, failures to repair DNA damage would not only affect the maintenance of self-renewal and differentiation capacity, but would also lead to genomic instability and pass genomic aberrations to the progeny [43]. The detailed DNA repair mechanisms of stem cells is still elusive. There are limited studies that compare the DNA repair mechanism between human ESCs or mouse ESCs and differentiated cells, and even fewer investigations of iPSCs and comparisons between them and ESCs or somatic cells. However, these limited studies have suggested a distinct DNA repair mechanism in stem cells. In hESCs, it was found that upon ATM inhibition, the formation of γH2AX foci induced by ionizing radiation was not completely inhibited, whereas upon inhibition of ATR, it was [44]. Moreover, another study showed that ATM−/− knockout hESCs incapable of mediating ATM-dependent phosphorylation of CHK2, p53 and γH2AX after irradiation did not display any genomic alterations or karyotypic abnormalities [45]. These results indicate that unlike somatic cells, hESCs do not rely exclusively on ATM-dependent mechanisms in DSB response and repair. There are also studies demonstrating that the relative contributions of DSB repair pathways were different between ESCs and somatic cells. When knocking out Rad54, a gene promoting homologous recombination (HR) by stimulating DNA pairing activity of Rad51, both Rad54−/− knockout mESCs and Rad54−/− adult mice were sensitive to the DNA cross-linking agent mitomycin C. In contrast, only Rad54−/− mESCs showed hypersensitivity to ionizing radiation, but Rad54−/− adult mice did not. As established, mitomycin C-induced DSBs are predominantly repaired by HR, whereas ionizing radiation-induced DSBs are repaired by NHEJ. This indicates that compared with adult mice, in mESCs, NHEJ plays a relatively minor role in DSBs repair. During mouse development, the relative contribution of HR and NHEJ to DSBs repair apparently shifts [46]. To repair DSBs, mESCs predominantly utilize HR (80%) with minimal contribution of NHEJ (20%), whereas the inverse was the case for mouse embryonic fibroblasts when the DNA repair capacities were directly measured by specific plasmid reactivation assays [47]. This result was supported by another study, which suggested that HR appeared to be the predominant pathway of choice to repair induced or spontaneous DNA damage throughout the mESC cycle in contrast to fibroblasts [48]. So far, only very few studies have addressed the DDR in iPSCs. Based on the available evidence, the response of iPSCs to DSBs might resemble that of hESCs [49,50]. Taken together, these studies clearly indicate distinct repair mechanisms for DSBs in embryonic/induced pluripotent stem cells as compared to other dividing somatic cell types. Our current results add new support to this.

As discussed above, FUS is known to participate in DDR and DNA repair through multiple ways, particularly in DNA DSB damage response and repair. However, different DNA damage repair pathways could not sufficiently explain the minor fraction of non-recruiting cells within the hiPSC population, as FUS reportedly participates in both pathways. In more detail, in cultured primary neuronal mouse models and immortalized cell models, FUS was required for both HR and NHEJ pathways. Knockdown of FUS reduced both pathways in these models [8,12]. Notably, all these previous studies about the role of FUS in DDR were limited to either cultured cancer lines such as HeLa, HEK293T, Neuro2, U2OS cells, etc., or in primary neuron cultures. To our knowledge, there is to date no DDR investigation on iPSCs and their derivates—the role of FUS remains elusive here. With this in mind, our current results suggest that the minor fraction of non-recruiting hiPSCs uses some pathways not involving FUS when responding to laser-induced DNA damage. Our observations call for further investigations of the DDR and repair pathways in hiPSCs to explain the occurrence of that distinct minor fraction that exhibits no FUS recruitment. For example, hiPSCs might feasibly switch from one repair mode to another when passing through the different phases of the cell cycle.

Apart from a minor hiPSC fraction, a gross FUS recruitment per se always occurred, but our study suggests that the fine-tuning of FUS dynamics in DDR depends on the particular cell type. Although it was difficult to find a common denominator of FUS kinetics in each cell type, we could narrow down the variations in kinetics mainly to the plateau and the dissociation phase. Typically, FUS-eGFP was quickly recruited to the DDS after laser irradiation independent of the cell type used, which was consistent with previous studies [8,12,16]. Our analysis suggests that FUS kinetics during the recruitment lag and association phase do not differ between the cell lines. Surprisingly, of particular importance for explaining a putative selective vulnerability of MNs might be the dissociation phase. The main measures of the dissociation phase differed markedly from other cell types, with a long dissociation time and a low *K_off_* value in the case of MNs. FUS has diverse cellular functions, including, e.g., transcriptional regulation [51,52] and stress granule dynamics [53]. If FUS disassembly takes particularly long in MNs, this could mean that FUS is tied up longer in DDR than in other cell types, possibly impairing its other cellular functions, specifically in MNs. Since transcriptional stalling and stress granule response are central in the pathophysiology of ALS, this might already be a significant disadvantage of MNs.

Potential limitations of the study include that we c-terminally tagged the FUS protein with the eGFP. Thus, further studies need to address whether n-terminally tagged FUS might behave differently. This might be the case since the NLS sequence is located c-terminally in FUS. Different linkers have been tested and proposed to be utilized as models of different severity of FUS pathologies. We decided to use a rather long linker (see Appendix A) so as to not artificially increase cytoplasmic FUS mislocalization. However, a long linker might affect FUS-eGFP nuclear egress since this was reported to mainly depend on passive diffusion [54]. We also did not systematically measure the FUS protein level prior to and after UV laser cutting in individual cells. 

Thus, against our a priori hypothesis, we did not find fundamental differences in cell type specificity of FUS protein behavior in DDR, which likely could explain the selective cell type vulnerability of motor neurons in the case of FUS mutations. Surprisingly, we found the most obvious phenotype in hiPSCs, which showed a total lack of FUS response to DDR in ~ one third of irradiated cells. The quantitative approach of recruitment kinetic analysis of DDR revealed slight but significant differences depending on the investigated cell model. Within that, FUS dissociation time but not association time was particularly long in MNs with a low *K_off_* value. Nevertheless, a minor kinetic difference in these upstream parameters could have drastic consequences for the complex repair machinery further downstream, especially in postmitotic cells. Whether these are sufficient to induce a selective vulnerability needs to be proven in future studies.

## 4. Methods and Materials

### 4.1. hiPSCs Culture

All gene-edited hiPSCs were generated previously in the group [25,26]. Clinical features of the donors are summarized in Table 1. In brief, healthy donor-derived fibroblasts were obtained and reprogrammed by the classical Yamanaka set, which comprises OCT4, SOX2, KLF4 and MYCC [20]. Both gene-unedited and -edited iPSCs were maintained on Matrigel (BD Bioscience, San Jose, CA, USA) coated 6-well plates with daily changed mTeSR-1 media (Stemcell Technologies, Vancouver, BC, Canada). Cells were routinely passaged when a 70–80% confluence was reached using ReLeSR (Stemcell Technologies) and following the manufacturer’s instructions. For staining and laser irradiation experiments, iPSCs were cultured on 4-well plates with cover slips and Fluorodishes (Thermo Fisher Scientific, Waltham, MA, USA), respectively.

### 4.2. Differentiation into Meso- and Endodermal Cells

iPSCs colonies were cultured under standard condition and detached with 1 mg/mL dispase (Thermo Fisher Scientific). Floating aggregates were transferred into ultra-low attachment plates (NUNC) in ES medium containing 78% KnockOut-DMEM (Thermo Fisher Scientific), 20% KnockOut serum replacement (Thermo Fisher Scientific), 1% MEM non-essential amino acids (Thermo Fisher Scientific), and 1% penicillin/streptomycin/glutamine (Thermo Fisher Scientific), supplied with 5 μM Y-27632 (Ascent Scientific, London, UK). On day 2, Y-27632 was omitted. On day 4, the formed embryoid bodies (EBs) were seeded onto gelatin (0.1%, Millipore, Burlington, MA, USA)-coated plates. The EBs were further differentiated for two weeks with meso-/endodermal differentiation medium that contains 76.9% DMEM (high glucose, Invitrogen, Waltham, MA, USA), 20% FCS (PAA), 1% penicillin/streptomycin/glutamine, 1% non-essential amino acids, 0.1% β-mercaptoethanol (Invitrogen) and 1% α–thioglycerol.

### 4.3. NPC Generation and Culture

NPC generation was performed as previously described [55]. Briefly, on day 0, iPSCs were detached using dispase (Thermo Fisher Scientific) and collected in ES medium supplied with 1 µM Dorsomorphin (Tocris Bioscience), 10 µM SB-431542 (Tocris Bioscience, Bristol, UK), 0.5 µM pumorphamine (PMA) (Cayman chemical company, Ann Arbor, MI, USA) and 3 µM CHIR99021 (Cayman chemical company). On day 2, culture medium was switched into N2B27 medium (all from Thermo Fisher Scientific) containing 48.75% Neurobasal Medium, 48.75% DMEM/F12, 1% penicillin/streptomycin/glutamine, 1% B27 supplement without Vitamin A, and 0.5% N2 supplement. The growth factors remained the same. On day 4, SB-431542 and Dorsormorphin were withdrawn, while 150 µM ascorbic acid (AA) (Sigma Aldrich, St. Louis, MO, USA) was added. On day 6, formed EBs were separated and plated on Matrigel-coated plates. For further cultivation, N2B27 medium was supplemented with 150 µM AA, 0.5 µM PMA and 3 µM CHIR99021. NPCs were passaged once they were confluent. Then, 10 min incubation with accutase (Sigma Aldrich) at 37 °C was used for detaching cells during passaging.

### 4.4. Differentiation of Human iPSCs to Striatal Neurons

Striatal neurons were directly differentiated from iPSCs [25]. Differentiation started when iPSCs population reached 80–90% confluence on a 6-well plate. Medium was changed to N2B27 medium. N2B27 medium was supplemented with different factors specific to the present differentiation phase: for the first 4 days (day 0–4), 10 μM SB431542, 1 μM Dorsomorphin and 1 μM IWP2 (Sigma), for days 6–8, 1 μM IWP2 and 0.2 μM Purmorphamin, day 10 without any supplement, and from day 12 onwards, 0.5 μM dibutyryl-cAMP (Sigma), 20 ng/mL BDNF (Promega, Madison, WI, USA),10 ng/mL GDNF (Sigma) and 1 ng/mL TGF-β3 (Peprotech, Rocky Hill, NJ, USA) were used. Medium was changed every other day. On day 20, cells were treated with pre-warmed accutase for 10 min. Detached cells were washed and collected in N2B27 medium. Supernatant was discarded after centrifuge for 5 min at 250 g. Cells were re-suspended in cultivation media and seeded onto Matrigel-coated 6-well plates with a density of 1.9 × 10^5^ cells/cm^2^. On day 30, cells were seeded onto poly-L-ornithine (15%, Sigma)/Laminin (1:100, Roche, Basel, Switzerland)-coated fluro-dishes using the accutase method described above. Then, 3 × 10^5^ cells/dish were seeded for UV laser micro-irradiation experiments. 

### 4.5. Differentiation of Human NPCs into Motor Neurons

The differentiation protocol was as described previously [39]. In brief, from day 0, NPCs were cultured in N2B27 medium containing 1 ng/mL BDNF, 0.2 mM AA, 1 µM RA, 1 ng/mL GDNF, and 0.5 μM SAG. Medium was changed every other day. On day 6, the growth factors changed to 0.1 mM dibutyryl-cAMP, 2 ng/mL BDNF, 1 ng/mL TGFβ-3, 2 ng/mL GDNF and 5 ng/mL Activin A (Biomol GmbH, Hamburg, Germany). From day 7, Activin A was omitted. On days 8 to 10, cells were passaged onto 15% PLO/Laminin-coated plates with accutase detaching method. Medium was changed every other day. On day 45, cells were used for the experiments.

### 4.6. Hela* Cell-Line (Gene-Edited)

HeLa FUS-eGFP cell line was kindly provided by Dr. Ina Poser/Prof. Anthony A. Hyman, Max Planck Institute for Molecular Cell Biology and Genetics, Dresden, Germany. The cell line uses a BAC system to overexpress FUS c-terminally tagged with eGFP [27].

HeLa cell lines were cultured in 6-well plates in HeLa medium containing 89% DMEM (high glucose), 10% FCS, 1% penicillin/streptomycin. Cells were passaged two times per week, when becoming confluent. This was carried out by first washing them with warm PBS, followed by incubation with 750 µL/well 0.05% Trypsin until cells started to detach. Enzymatic reaction was stopped by the addition of 1.5 mL HeLa medium. Cells were then centrifuged at 250× *g* at rt for 5 min. Supernatant was aspirated and the pellet resuspended in fresh HeLa medium. Cells were split at rations between 1:20 and 1:50.

### 4.7. CRISPR/Cas9 Genome Editing

To obtain FUS protein in the iPSCs labeled with eGFP, the gene was edited by the CRISPR/Cas9 method [56]. This protocol has been published previously. Briefly, using Fugene HD^®^ (Promega), iPSCs were transiently transfected with two plasmids. The pX335B vector, containing Cas9n, was provided by the laboratory of Dr. Boris Greber, (Max Planck Institute for Molecular Biomedicine, 48149 Münster, Germany, Appendix A) [57]. The second plasmid (pEX-K4) was used as a template for homology directed repair (Eurofins Genomics, Appendix A). The targets for the double nicking approach are as follows: T1—gcgagtatcttatctcaagt; T2—gttaggtaggaggggcagat. The repair construct contained our desired mutation, a short linker, eGFP, and was flanked by homology arms covering 500 bp upstream and 400 bp downstream of the induced double nick. At 24 h after transfection, cells were directly selected by 0.4 μg/μL Puromycin (InvivoGen, San Diego, CA, USA). Then, surviving cells were cultured for recovery, and about 3–7 days later, could be passaged onto a new plate (at 2000 cells per 6-well). About 10–14 days after the last passage, eGFP-positive colonies emerged and were selected according to their green fluorescence. During the whole procedure, the iPSCs were cultured with the TeSR-E8 medium. To confirm the genotype of the FUS-eGFP-positive iPSCs after CRISPR/Cas9 editing, a PCR was run on the genomic DNA that was isolated (DNeasy Blood and Tissue Kit, Qiagen, Germantown, MD, USA) from gene-edited iPSCs following the manufacturer’s instructions. A forward primer (CAGTTGAACAGAGGCCATAGG) targeting upstream endogenous C-terminal ending, and a reverse primer (CTCTCTACCTTCCTGATCGGG) targeting downstream eGFP were used. Two PCR fragments (1257 bp for FUS with EGFP and 528 bp for FUS without EGFP) can be produced. gDNA from non-transfected iPSCs was amplified as negative control (528 bp for FUS without EGFP).

### 4.8. Immunofluorescence Stainings

Cells were washed with DPBS (LifeTechnologies, Carlsbad, CA, USA) twice and fixed by 4% PFA (Electron Microscopy Sciences, Hatfield, PA, USA) at room temperature for 15 min. After fixation, the PFA was aspirated and cells washed three times with DPBS. Fixed cells were then permeabilized with 0.2% Triton X-100 (in DPBS, Thermo Fisher Scientific) for 10 min. Samples were then incubated with blocking solution (1% BSA, 5% donkey serum, 0.3 M glycine and 0.02% Triton X in PBS) at room temperature for one hour. Following blocking, the samples were incubated with primary antibodies diluted in blocking solution at 4 °C overnight. The next day, the sample was washed three times with DPBS, 5 min each time. Samples were then incubated with secondary antibodies diluted in blocking solution at room temperature for one hour while avoiding exposure to light. After the secondary incubation, samples were washed three times with DPBS. An amount of 0.75 μL/mL Hoechst33342 (LifeTechnologies) was used as a counterstain at room temperature for 3 min. Samples were again washed with DPBS three times. The following primary antibodies were used: mouse anti-FUS (1:3000, Sigma), chicken anti-GFP (1:500, Sigma). The following secondary antibodies were used: donkey anti-mouse Alexa Fluor 594 (1:2000, Thermo Fisher Scientific) and donkey anti-chicken Alexa Fluor 647 (1:2000, Thermo Fisher Scientific).

### 4.9. Laser Irradiation-Induced DNA Damage Assay

All different cell types and lines were obtained, cultured and, if applicable, differentiated as described above. To perform the DNA damage laser irradiation assay, a final split was performed on all cell types to obtain cell cultures in the final assay format in 3.5 cm dishes (FluoroDish with 160 µm cover glass bottom, World Precision Instruments, Sarasota, FL, USA) at 3 × 10^5^ cells per dish. All subsequent imaging of DNA damage response to laser irradiation sites was performed as described previously. In brief, a focused 355 nm UV laser beam was directed through a stereotactic galvanometric mirror box to desired x-y-z positions in cell samples held on a standard inverted Axio Observer Z1 Zeiss microscope equipped with a motorized stage and a piezo-electric Z-actuator. A Zeiss alpha Plan-Fluar 100 × 1.45 oil immersion objective was used, and 24 laser shots in 0.5 μm steps were administered over 12 μm linear cuts located within cell nuclei. The cellular response to this DNA damage comprised a fast recruitment of FUS-eGFP to the laser cut site followed by its slower withdrawal (on–off kinetics) and was recorded live over 35 min by confocal spinning disc imaging of the eGFP tag using a 488 nm laser line and a 12-bit Andor iXON 897 EMCCD camera (512 × 512, 16 μm pixels, 229.55 nm/pixel at 100× magnification) at initial 1 fps and later 0.2 fps during the slower withdrawal phase. All image frames of the obtained movie stacks presented in this report (Figure 2b) are shown in the look-up-table (LUT) “Green Fire Blue” provided in the FIJI software Version 1.48a and 1.52b), (i.e., low eGFP intensities are shown in blue and high intensities in green shades.

### 4.10. Analysis Method of FUS Kinetics

For analysis of FUS kinetics, maximum intensity projections of the Z-stacks were first generated in Fiji software (Version 1.48a and 1.52b) manually for each video. Then, a serial of nuclei detection, laser irradiating site detection, and intensity measurement of areas of interest were performed with a sequence of commands in FIJI software (Version 1.48a and 1.52b) executed with a macro. (For the macro code see: https://github.com/Glyphus/Fiji-macros/blob/main/UV-Laser_ablation.ijm (accessed on 28 February 2024))

These macros provided result in tables listing the intensity values of both laser irradiating sites and their respective background (Figure 3). They were further organized by KNIME to automatically obtain the normalized intensity change for each FUS recruited laser irradiating site. For the kinetics analysis, important time points, the duration of each phase and the constants were calculated by an R-script. (For the R-script see: https://github.com/Glyphus/R-Scripts/blob/main/UV-Laser_ablation_evaluation.R (accessed on 28 February 2024)).

The results were presented both in a curve figure and text (Figure 4).

## Figures and Tables

**Figure 1 ijms-25-03526-f001:**
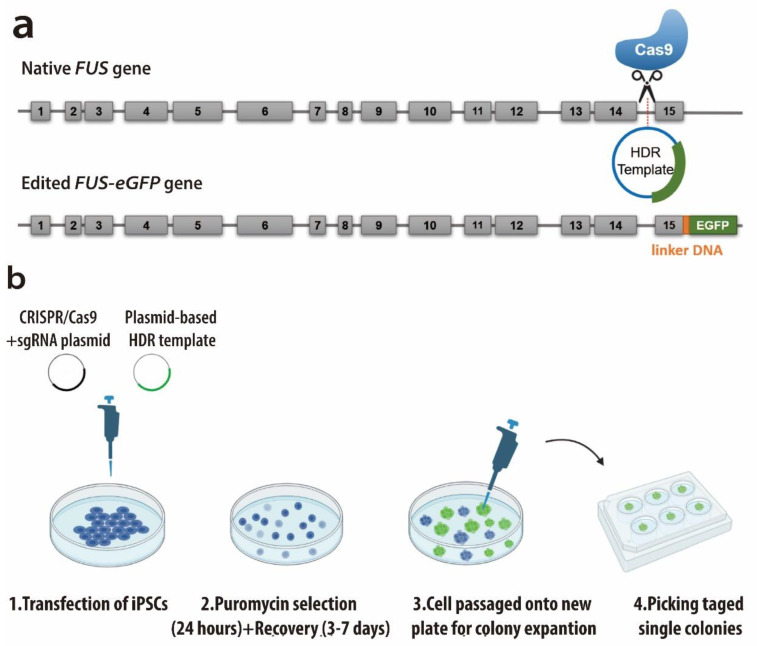
Generation of gene-edited wild-type FUS-eGFP hiPSCs. (**a**) Schematic representation of *FUS* gene with numbered exons, upper panel: before gene editing, Cas9n cutting site was shown; lower panel: after gene editing. (**b**) Schematic of CRISPR-Cas9n-mediated gene editing procedure. (**c**) Immunofluorescence staining of FUS-antibody (Ab), GFP-antibody, and Hoechst, representative images show a perfect match of FUS-eGFP signal with both FUS and GFP antibody stainings, bar: 50 μm. (**d**) Genotyping PCR of gene-edited hiPSC lines. Shown is a representative line used in the study. (**e**) Western blot of iPSCs prior to and after successful gene editing. Shown is a representative line used in the study. Full dataset is shown in Appendix A. Heterozygous cell lines were picked up for later investigation.

**Figure 2 ijms-25-03526-f002:**
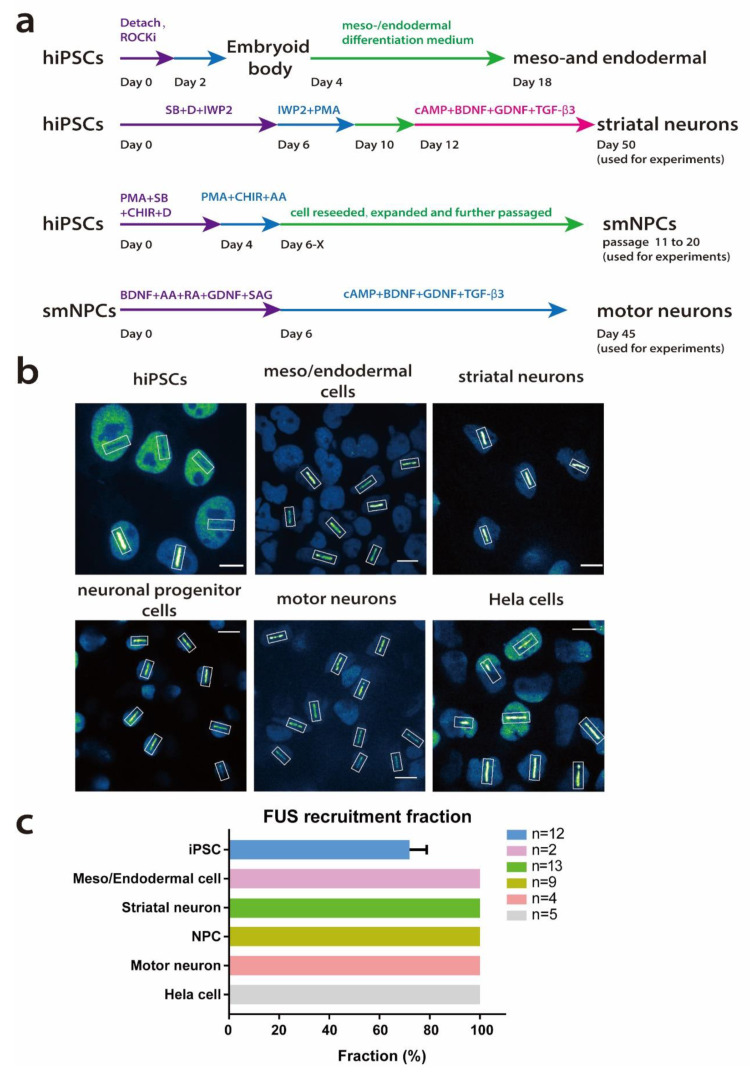
Wild-type FUS protein response to UV laser irradiation-induced DNA damage. (**a**) Differentiation scheme of different cellular models. (**b**) Representative images of each cell type. FUS-GFP was imaged in live cell microscopy. Of note, all cells expressed FUS protein in the nucleus. All cells showed recruitment of FUS-eGFP to DDS except a fraction of undifferentiated hiPSC. Laser irradiating sites were marked with white rectangles. Images are shown false colored with Fiji “Green Fire Blue” LUT, Scale Bars = 10 μm. (**c**) Fraction of FUS-eGFP-recruiting cells in different cellular models. Each n represents one independent experiment. At least 5 videos which recorded 2–10 cells per video were irradiated and analyzed. Error bars = SEM. “n” represents biological replicates.

**Figure 3 ijms-25-03526-f003:**
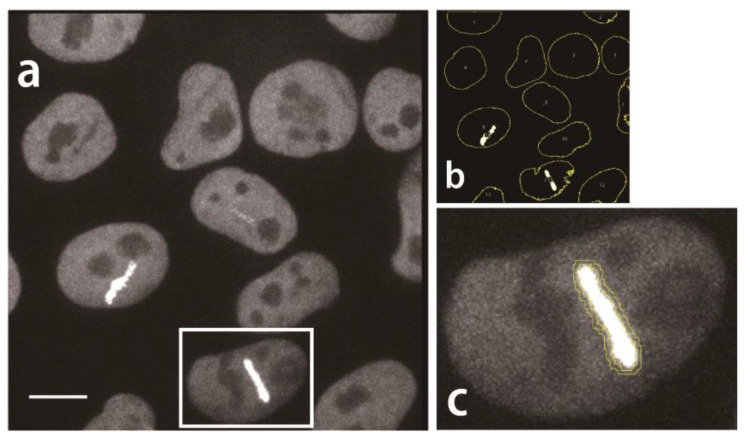
Detection of FUS recruitment at laser irradiating sites with Fiji. (**a**) Representative maximum intensity projection image of FUS recruitment at laser-induced DNA damage sites in hiPSCs. Nuclei were visible as FUS-eGFP mainly localized to the nucleus (see also Figure 1). Bright strips in the nuclei depict FUS-GFP recruited to DDS after a 10 µm linear laser irradiation was applied to induce DNA damage. Scale bar = 10 μm. (**b**) Automatic detection of nuclei and FUS recruitments by our custom-tailored Fiji macro script. (**c**) Zoomed in nucleus from Figure 3a. FUS-eGFP recruitment and background detection was conducted for each nucleus individually. After threshold detection of the FUS-eGPF recruitment site, a two-pixel-wide ring surrounding the DDS was generated and used to determine background fluorescence near the DDS. Images are shown false colored with Fiji “Grays” LUT.

**Figure 4 ijms-25-03526-f004:**
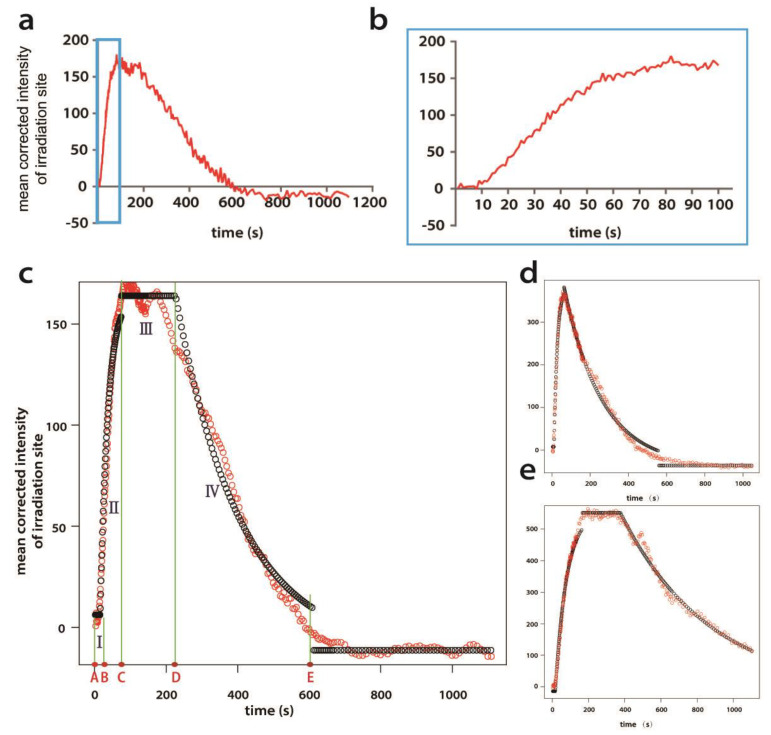
Method of kinetic analysis of FUS-eGFP recruitment dynamics at DDSs. (**a**) Representative graph of mean background-corrected intensity change at laser-irradiated sites over time. Laser irradiation was applied to the nucleus at time point 1 s. Rectangle area indicates the first 100 s. (**b**) Magnification of the first 100 s of (**a**). FUS-eGFP recruitment lags ~10 s behind the laser irradiation (**c**–**e**). Representative graphs generated by our R-script, showing different examples of measured FUS-eGFP intensity over time. Red circles represent measured data, and black circles represent robust fitting of our model. (**c**) FUS-eGFP recruitment showed a plateau and fluorescence reaches pre-irradiating level within the 10 min imaging window. Four phases are defined during the recruitment of FUS-eGFP: I recruitment lag phase, II association phase, III plateau phase, and IV dissociation phase. Five time points detected for the start and end points of each phase: time point A representing when laser irradiation was applied to the nucleus, point B representing the beginning of FUS-eGFP recruitment to the DDS, point C representing end of recruitment, point D representing start of FUS-eGFP dissociation from the DDS, and point E representing end of FUS-eGFP dissociation. (**d**) FUS-eGFP recruitment without a plateau. (**e**) The intensity of FUS-eGFP fluorescence did not reach the pre-irradiating level within the measured time window.

**Figure 5 ijms-25-03526-f005:**
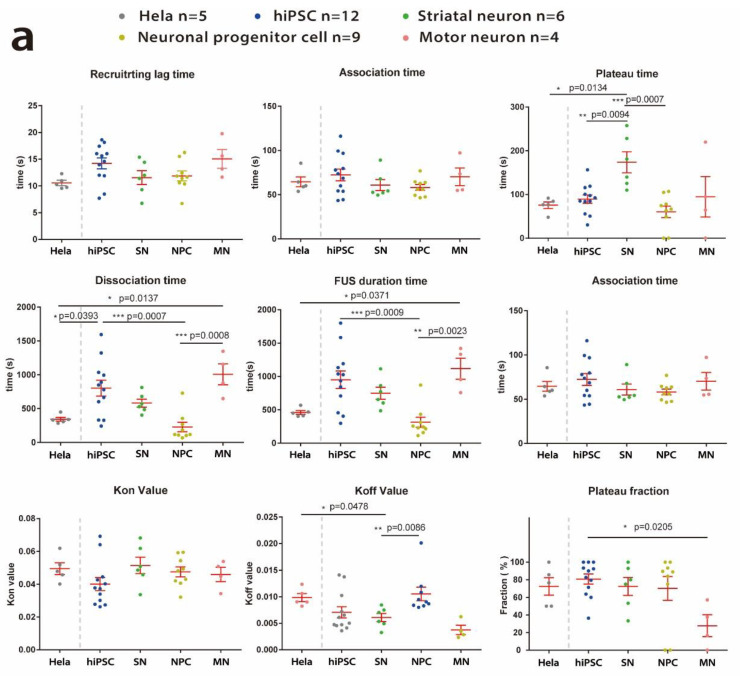
Kinetics analysis of wild-type FUS protein in different types of in vitro cellular models upon laser-irradiation-induced DNA damage. (**a**) Comparisons between Hela cell, hiPSC, striatal neuron, neuronal progenitor cell and motor neuron. One-way ANOVA followed by Tukey post hoc tests were used as statistical analysis, suggesting several significant differences in some parameters between types of cells. (**b**) Comparisons among the isogenic lines, Line Kolf Cas+—iPSC, Line Kolf Cas+—NPC and Line Kolf Cas+—motor neuron. One-way ANOVA analysis followed by Tukey post hoc tests suggested significant difference of some parameters. n represents experimental replicate. Each single experiment includes at least five videos, which recorded 2–10 laser-irradiated cells per video. Error bars represent the SEM; * indicates a significant difference within the group; *, **, *** represent *p*-values of 0.05, 0.01, and 0.001, respectively.

**Table 1 ijms-25-03526-t001:** Patient/proband characteristics.

		Sex	Age at Biopsy (Years)	Comment	Citation
hiPSC					
	iPSC 1	male	36	Donor 1, healthy volunteer	[25]
	iPSC 2.1	male	36	Donor 2, healthy volunteer	[25]
	iPSC 2.2	male	36	Different clone of Donor 2	[25]
	iPSC Kolf	male	55–59	KOLF, characterized in Wellcome Trust Sanger Institute, Hinxton, United Kingdom	Welcome Trust Sanger Institute; Hinxton; UK
HeLa					
	HeLa-FUS-eGFP	female	30		[27,28]

## Data Availability

All data are presented in this manuscript and Appendix A.

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
