# Peer review of "Cell-Type-Dependent Recruitment Dynamics of FUS Protein at Laser-Induced DNA Damage Sites"

_ijms, 2024, doi:10.3390/ijms25063526_

Round 1

Reviewer 1 Report

Comments and Suggestions for Authors

The authors address an interesting and important topic of cell type-specific susceptibility to some inherited pathologies. One of the functions of FUS is to participate in the DNA damage response. The authors investigated FUS recruitment to UV-irradiated sites in the nucleus in different cell types obtained by IPSC differentiation. Genetic modification of the FUS gene allowed lifetime visualization of the recruitment dynamics of the protein fused with eGFP to DNA damage sites. The researchers also present an algorithm for quantitative analysis of FUS kinetics. The study is relevant and technically sound. However, some comments should be addressed.

1. Since the authors discuss differences in FUS recruitment in different cell types, it is important to demonstrate the expression of cell type specific markers in the obtained cell population (either by immunofluorescence or RT-PCR).

2. It would be better to show the map of the plasmid used as a template for homology recombination to introduce eGFP into the genome of hiPSC. There is a confusion - in the Results the authors state that "wild-type FUS is tagged with a c-terminal eGFP", but in the Materials and Methods there is a mention of a "point mutation".

3. The main question of the article - the susceptibility of motor neurons to the FUS mutation in ALS - is poorly discussed. However, some of the parameters evaluated showed significant differences between motor neurons and other cell types (e.g. dissociation constant). Or is the repair-related function of FUS not crucial for the development of ALS?

Author Response

Comments and Suggestions for Authors

The authors address an interesting and important topic of cell type-specific susceptibility to some inherited pathologies. One of the functions of FUS is to participate in the DNA damage response. The authors investigated FUS recruitment to UV-irradiated sites in the nucleus in different cell types obtained by IPSC differentiation. Genetic modification of the FUS gene allowed lifetime visualization of the recruitment dynamics of the protein fused with eGFP to DNA damage sites. The researchers also present an algorithm for quantitative analysis of FUS kinetics. The study is relevant and technically sound. However, some comments should be addressed.

Response: We deeply acknowledge this very positive overall review and tried hard to address all the remaining comments.

  1. Since the authors discuss differences in FUS recruitment in different cell types, it is important to demonstrate the expression of cell type specific markers in the obtained cell population (either by immunofluorescence or RT-PCR).

Response: Our apologies if we were not clear enough. See also reviewer #2. We indeed used well established protocols in the lab of meso/endo/ectoderm differentiation (=Embryoid body formation), MSN and MN differentiation, which had been published a couple of times already including proof of fate and so on (e.g. doi: 10.1523/JNEUROSCI.0456-16.2016; doi: 10.1016/j.mcn.2018.08.002; doi: 10.3390/ijms21051797; doi: 10.1016/j.nbd.2015.07.017; doi: 10.1016/j.stemcr.2017.12.018; doi: 10.1038/s41467-017-02299-1). Nevertheless, we now present representative images in the supplement of the revised manuscript.

  1. It would be better to show the map of the plasmid used as a template for homology recombination to introduce eGFP into the genome of hiPSC. There is a confusion - in the Results the authors state that "wild-type FUS is tagged with a c-terminal eGFP", but in the Materials and Methods there is a mention of a "point mutation".

Response: We show the respective plasmid in the supplement of the revised manuscript. Our apologies for the mistake with the point mutation, this was a fault. This was corrected in the revised version of the manuscript.

  1. The main question of the article - the susceptibility of motor neurons to the FUS mutation in ALS - is poorly discussed. However, some of the parameters evaluated showed significant differences between motor neurons and other cell types (e.g. dissociation constant). Or is the repair-related function of FUS not crucial for the development of ALS?

Response: We very much appreciate this comment and add a respective paragraph at the end of the discussion section of the revised manuscript.

Reviewer 2 Report

Comments and Suggestions for Authors

The work by Niu and colleagues reports the generation of hiPSCs expressing EGFP-tagged FUS, by CRISPR-Cas9-mediated editing of one of the endogenous FUS loci. These cells were then differentiated in four different cell types to investigate FUS recruitment kinetics to DNA damage sites in different cells. To do so the authors performed laser micro-irradiation and live cell imaging experiments both in hiPSCs and hiPSC-derived cells as wells as in HeLa cells as control. The different cell models were compared based on a set of fixed parameters extrapolated from the recruitment curves, showing similar kinetics in all conditions.

Overall, the idea behind the study is interesting because aims at highlighting cell type-specific differences in the behavior of FUS protein during DNA damage response. This aspect is extremely relevant because DNA damage response is altered in the neurodegenerative disorder ALS and FUS mutations are associated with this same disease that affects only motor neurons.

On the other side, laser irradiation to study the recruitment of proteins of interest to sites of DNA damage is a well established techniques and several protocols have been developed and published (doi:10.1016/j.dnarep.2023.103545, doi:10.3791/56213, doi:10.1016/j.xpro.2022.101146), therefore this approach is not novel and the frequent statements of novelty should be removed.

Detailed comments below.

Mayor points:

1.     Figure 1c. It is not very informative to perform a double staining with anti-FUS and anti-EGFP on cells expressing FUS-GFP. I would rather suggest to provide image of control cells (not modified through CRISPR/Cas9) stained with anti-FUS and anti-EGFP to confirm that the EGFP signal is specific of your modified cells and that endogenous untagged FUS has the same localization of FUS-EGFP.

2.     Figure 1d. Lanes have different dimensions and it is difficult to compare them with the marker. Please provide uncut original images as supplementary materials to show the entire gel (at least one in which it is possible to see marker, positive and negative cells.

3.     How do the authors validate the different cell lines obtained after differentiation?

4.     The analysis protocol should be corrected according to published guidelines for the analysis of this type of experiment (see for example doi:10.1016/j.dnarep.2023.103545). For example, the authors should subtract the background (cell-free) signal from both the irradiated region and a control one. The control region (which accounts for photo-bleaching) needs to be either the entire nucleus or a region distant from the irradiated one. This because other proteins have a more disperse recruitment, so the approach proposed by the authors (using as background the region around the irradiation) is not suitable for all proteins.

5.     The authors used clones with constant and endogenous FUS expression levels. However, from Figure 4 it seems that FUS maximum recruitment varies among cells, is this so? How variable was the basal and maximum fluorescence after irradiation? I am a bit concerned about this aspect because FUS is strongly recruited to sites of irradiation, causing the signal to reach saturation. The plateau that the authors observed in some cells could derive from this.

6.     Figure 5. The authors decided to compare different cells based on the calculated parameters but I would suggest to provide also the recruitment curves (as those in Figure 4), which are easily visualized and immediately understandable by the reader. In addition, please provide quantification of basal pre-irradiation fluorescence and maximum fluorescence at irradiated sites for all different cell lines. This is extremely important because different expression levels could account for different recruitment kinetics.

7.     The protocol for laser irradiation is missing (Methods section).

Minor points:

1.     L58-60 “In this respect, FUS was reported to play a role in the two major pathways of double-strand break (DSB) repair, namely in the homologous recombination (HR) and non-homologous end joining (NHEJ)”. It seems to me that reference 9 is not appropriate in this point.

2.     I would add a reference on the cell type-specific susceptibility in ALS (for example doi: 10.3389/fnins.2019.00532)

3.     L122-123. Please add a reference for the phrase “FUS-ALS causing mutations impair DDR, including proper FUS recruitment to UV laser-induced DDS”.

4.     Figure 2b. Nuclear signal covers the FUS-GFP one, I would suggest showing only the latter.

5.     Figure 2b. hiPSCs that do not recruit FUS-GFP show instead a release, please comment this aspect.

6.     L285-292. The issue with this type of approach is not to keep GFP fluorescent but to avoid disturbing the POI’s function. It is possible to optimize the fusion strategy by choosing the N- or C-terminal and by adding or changing a linker sequence between the POI and the fluorescent one. In addition, what do the authors mean by saying that GFP ameliorate pathological hallmark?

7.     The macro and script as they are presented are difficult to be used by another user. I would suggest to make them available for the community on the ImageJ portal and then sharing the https.

Comments on the Quality of English Language

I would suggest another check of the English from a native speaker. The text is perfectly understandable but could be improved. 

Author Response

The work by Niu and colleagues reports the generation of hiPSCs expressing EGFP-tagged FUS, by CRISPR-Cas9-mediated editing of one of the endogenous FUS loci. These cells were then differentiated in four different cell types to investigate FUS recruitment kinetics to DNA damage sites in different cells. To do so the authors performed laser micro-irradiation and live cell imaging experiments both in hiPSCs and hiPSC-derived cells as wells as in HeLa cells as control. The different cell models were compared based on a set of fixed parameters extrapolated from the recruitment curves, showing similar kinetics in all conditions.

Overall, the idea behind the study is interesting because aims at highlighting cell type-specific differences in the behavior of FUS protein during DNA damage response. This aspect is extremely relevant because DNA damage response is altered in the neurodegenerative disorder ALS and FUS mutations are associated with this same disease that affects only motor neurons.

Response: We deeply acknowledge this overall positive review!

On the other side, laser irradiation to study the recruitment of proteins of interest to sites of DNA damage is a well established technique and several protocols have been developed and published (doi:10.1016/j.dnarep.2023.103545, doi:10.3791/56213, doi:10.1016/j.xpro.2022.101146), therefore this approach is not novel and the frequent statements of novelty should be removed.

Response: We agree with the reviewer that the technology to study protein recruitment at DNA damage sides induced by laser irradiation is not novel. We also included the important literature mentioned by the reviewer and discussed it in much more detail in the revised version. Of note, however, we do not present guidelines on how to perform laser irradiation experiments, but rather an algorithm to evaluate them including making the respective Fiji macro and R script openly available. We tried hard to stick to the mentioned guidelines and are convinced that we did so, which we hope to make more clear in the revised version of the manuscript. Nevertheless, we additionally toned down this statement.

Mayor points:

  1. Figure 1c. It is not very informative to perform a double staining with anti-FUS and anti-EGFP on cells expressing FUS-GFP. I would rather suggest to provide image of control cells (not modified through CRISPR/Cas9) stained with anti-FUS and anti-EGFP to confirm that the EGFP signal is specific of your modified cells and that endogenous untagged FUS has the same localization of FUS-EGFP.

Response: Our intention was to show that the GFP signal overlaps with FUS antibody staining, to show that the GFP is properly linked to FUS and the expression pattern (mainly nuclear localization) is not disturbed. We do understand the reviewer’s point and added data on this using western blots of cell lines prior and post CRIPSR/Cas tagging in the revised Figure 1d and in the supplement.

  1. Figure 1d. Lanes have different dimensions and it is difficult to compare them with the marker. Please provide uncut original images as supplementary materials to show the entire gel (at least one in which it is possible to see marker, positive and negative cells).

Response: We inserted a novel figure 1d and showing one in which it is possible to see marker, positive and negative cells, the entire gel is provided in the supplement

  1. How do the authors validate the different cell lines obtained after differentiation?

Response: Our apologies if we were not clear enough. See also reviewer #1. We indeed used well established protocols in the lab of meso/endo/ectoderm differentiation (=Eb body formation), MSN and MN differentiation, which had been published a couple of times already including proof of fate and so on (e.g. doi: 10.1523/JNEUROSCI.0456-16.2016; doi: 10.1016/j.mcn.2018.08.002; doi: 10.3390/ijms21051797; doi: 10.1016/j.nbd.2015.07.017; doi: 10.1016/j.stemcr.2017.12.018; doi: 10.1038/s41467-017-02299-1). Nevertheless, we now present representative images in the supplement of the revised manuscript.

  1. The analysis protocol should be corrected according to published guidelines for the analysis of this type of experiment (see for example doi:10.1016/j.dnarep.2023.103545). For example, the authors should subtract the background (cell-free) signal from both the irradiated region and a control one. The control region (which accounts for photo-bleaching) needs to be either the entire nucleus or a region distant from the irradiated one. This because other proteins have a more disperse recruitment, so the approach proposed by the authors (using as background the region around the irradiation) is not suitable for all proteins.

Response: We appreciate the concern of the reviewer for the fidelity of our analysis and think that after careful study of the proposed guidelines, we do not only match the criteria presented in doi:10.1016/j.dnarep.2023.103545 but additionally corrected for other experimental systematic errors. First of all, we do not only take single images at each time point but stacks – this accounts for laser recruitment events in the axial dimension. Furthermore, the achieved framerate in our setup allows us to quantitate the dynamics of the recruitment and dissociation. The time interval for imaging FUS recruitment as suggested in doi.org/10.1016/j.xpro.2022.101146 is 30 s – this would have been too slow to determine the kinetic parameters for FUS accurately.

After image acquisition, we perform a stack registration, which accounts for lateral shift of the nucleus/ irradiation site and therefore minimizes the readout window used for the temporal intensity measurement. Bleaching correction is performed not only on the nucleus but the whole view field minus the images, where the UV irradiation is performed – since these are presented as outliers when performing an exponential bleaching correction. The peri-irradiated region is to our understanding the best background correction region, at least for your protein of interest, because it is precisely the region where FUS is recruited from. The recruitment is happening locally. Molecules that diffuse from distal nuclear regions through the peri-irradiated area are detected with this method, since they are stabilizing the fluorescence intensity, which would otherwise decrease because of the constant recruitment from this region. A disperse recruitment, as mentioned by the reviewer, is easily detected with our workflow. Since we do not rely on an arbitrary region of interest, but robust thresholding, using published algorithms, the likelihood is very high that we will detect recruitment that is significantly more intense than the nucleus within a selected ROI. Since we use a maximum intensity projection for the thresholding, a bona fide detected region of interest will contain a recruitment event throughout the entire video.

One downside to this approach is, however, that we cannot account for the amount of fluorophore photo bleached during the UV irradiation. Hence this approach is not suitable for FRAP experiments or other setups, where a measure of an immobile fraction needs to be taken. The photo bleaching during the UV irradiation does not affect the kinetics of the protein recruitment though, since these parameters are all assessed after bleaching event has occurred. The bleaching happening throughout the image acquisition, which will change the kinetic parameters, is corrected for as explained above.

  1. The authors used clones with constant and endogenous FUS expression levels. However, from Figure 4 it seems that FUS maximum recruitment varies among cells, is this so? How variable was the basal and maximum fluorescence after irradiation? I am a bit concerned about this aspect because FUS is strongly recruited to sites of irradiation, causing the signal to reach saturation. The plateau that the authors observed in some cells could derive from this.

Response: We thank the reviewer for his keen observation. The images presented in figure 2 and 3 used a modified LUT/ histogram for better graphical representation. We can assure the reviewer, that during image acquisition we never reached optical saturation. Even though we used clonal cell lines, FUS expression level might vary from cell to cell. We tried to account for it by measuring at least least 5 videos with at least 2-10 cells per biological replicate and only irradiated cells that did not show significantly higher levels of FUS-eGFP as seen in the live image of the microscope. We did, however, not systematically analyze the basal and maximum fluorescence.

  1. Figure 5. The authors decided to compare different cells based on the calculated parameters but I would suggest to provide also the recruitment curves (as those in Figure 4), which are easily visualized and immediately understandable by the reader. In addition, please provide quantification of basal pre-irradiation fluorescence and maximum fluorescence at irradiated sites for all different cell lines. This is extremely important because different expression levels could account for different recruitment kinetics.

Response: We agree with the reviewer, that recruitment curves can be easily interpreted by the reader and provided some as proof of principle as well as illustration of the result of our pipeline in figure 4. However, due to the sheer amount of conditions, biological replicates and cells investigated in this study a comprehensive installment would comprise more than 500 curves. We are afraid that blotting all curves of a single condition together in one graph will also be confusing to the reader. Furthermore, some parameters – like the koff value – are, albeit significantly, not different by a large amount and due to the exponential nature can be poorly judged by eye.

We did not systematically analyze the basal and maximum fluorescence level of each single cell. Even though we used clonal cell lines, FUS expression level might vary from cell to cell. We tried to account for it by measuring at least 5 videos with at least 2-10 cells per biological replicate and only irradiated cells that did not show significantly higher levels of FUS-eGFP as seen in the live image of the microscope.

  1. The protocol for laser irradiation is missing (Methods section).

Response: Our apologies, this is added in the revised version now.

Minor points:

  1. L58-60 “In this respect, FUS was reported to play a role in the two major pathways of double-strand break (DSB) repair, namely in the homologous recombination (HR) and non-homologous end joining (NHEJ)”. It seems to me that reference 9 is not appropriate in this point.

Response: This is corrected in the revised version now.

  1. I would add a reference on the cell type-specific susceptibility in ALS (for example doi: 10.3389/fnins.2019.00532)

Response: This is added in the revised version now.

  1. L122-123. Please add a reference for the phrase “FUS-ALS causing mutations impair DDR, including proper FUS recruitment to UV laser-induced DDS”.

Response: This is added in the revised version now.

  1. Figure 2b. Nuclear signal covers the FUS-GFP one, I would suggest showing only the latter.

      Response: We are afraid that there is a misunderstanding of the images in Figure 2. We only present the (FUS)-eGFP channel. We did all experiments without addition of Hoechst33342 or any similar counterstain. Since FUS is a nuclear protein, this is enough to detect nuclei. For better visualization, the images in figure 2b are presented with the look-up-table (LUT) “Green Fire Blue”. Low eGFP intensity is shown as blue and high intensity as green/whitish. For reference, an unmodified LUT is presented in figure 3. We understand that the blue nuclear signal can be misinterpreted as nuclear counterstain and added a note about the LUT in the figure legend.

  1. Figure 2b. hiPSCs that do not recruit FUS-GFP show instead a release, please comment this aspect.

Response: We think that the release mentioned by the reviewer refers to the bright eGFP signal in the nucleus in the iPSC, which did not show recruitment. This is, however, a misinterpretation due to the LUT “Green Fire Blue”. FUS was not released into the nucleus in these cases but the nucleus itself had an overall bright eGFP signal, whereas the region of the UV laser was devoid of FUS. This was independent of FUS expression in the nucleus (depicted by this LUT depiction).

  1. L285-292. The issue with this type of approach is not to keep GFP fluorescent but to avoid disturbing the POI’s function. It is possible to optimize the fusion strategy by choosing the N- or C-terminal and by adding or changing a linker sequence between the POI and the fluorescent one. In addition, what do the authors mean by saying that GFP ameliorate pathological hallmark?

Response: We agree with the reviewer that protein tagging especially with large tags like GFP poses many challenged including the possibility to destroy the POI’s function. The tagging strategy we employed is well studied and published – both as overexpression system (https//doi.org/10.1038/nmeth.1199) as well as CRISPR engineered cell line (https://doi.org/10.1016/j.stemcr.2017.12.018). The comparison of N- vs. C-terminal fused eGFP is of interested, but beyond the scope of this manuscript. We observed in our laboratory that FUS tagged with eGFP shows a lower cytoplasmic mislocalization (unpublished data). According to Ederle et al. (https://doi.org/10.1038/s41598-018-25007-5) the NES of FUS is non- functional and nuclear export is mediated by passive diffusion. We do believe that addition of eGFP is interfering with the passive diffusion and therefore lowers the cytoplasmic mislocalization and ameliorates this pathological hallmark. Different linkers have been tested by (https://doi.org/10.1016/j.stemcr.2017.12.018) and proposed to be utilized as models of different severity of FUS pathologies. Details to your construct can be seen in the novel supplemental figure 1.

  1. The macro and script as they are presented are difficult to be used by another user. I would suggest to make them available for the community on the ImageJ portal and then sharing the https.

Response: We thank the reviewer for this suggestion and made the macro and R code available on github and provide the links in the revised version of the manuscript. This will encourage the scientific community to use and optimize our code as well as allow us to keep it up to date for newer versions of ImageJ and R.